# Sustained Virological Response Is the Most Effective in Preventing Hepatocellular Carcinoma Recurrence after Curative Treatment in Hepatitis C Virus-Positive Patients: A Study Using Decision Tree Analysis

**Kenji Imai \*, Koji Takai, Shinji Unome, Takao Miwa** ⓘ**, Toshihide Maeda, Tatsunori Hanai** ⓘ**, Yohei Shirakami, Atsushi Suetsugu and Masahito Shimizu** ⓘ

Department of Gastroenterology/Internal Medicine, Gifu University Graduate School of Medicine, 1-1 Yanagido, Gifu 501-1194, Japan; koz@gifu-u.ac.jp (K.T.); unome-shinji@hotmail.com (S.U.); takao.miwa0505@gmail.com (T.M.); toshi_z218@yahoo.co.jp (T.M.); hanai0606@yahoo.co.jp (T.H.); ys2443@gifu-u.ac.jp (Y.S.); asue@gifu-u.ac.jp (A.S.); shimim@gifu-u.ac.jp (M.S.)

**\*** Correspondence: ikenji@gifu-u.ac.jp; Tel.: +81-(58)-230-6308; Fax: +81-(58)-230-6310

**Abstract:** This study evaluated the factors that affect the recurrence of hepatocellular carcinoma (HCC) in hepatitis C virus (HCV)-positive patients, who had received curative treatment for initial HCC, using decision tree analysis in 111 curative cases. The enrolled patients were divided into three groups by the decision tree analysis as follows: Patients who achieved sustained virological response (SVR) after curative treatment belonged to Group 1 (*n* = 33), those who did not achieve SVR and with alpha-fetoprotein (AFP) levels < 11 ng/mL belonged to Group 2 (*n* = 30), and those who did not achieve SVR and with AFP levels ≥ 11 ng/mL belonged to Group 3 (*n* = 48). The Kaplan–Meier method revealed that Group 1 had significantly longer recurrence-free survival than Group 2 or 3 (*p* = 0.004). Moreover, there was no significant difference between patients achieving SVR with direct-acting antivirals and interferon therapy (*p* = 0.251). Group 3 had significantly poorer recurrence-free survival than Group 2 (*p* < 0.001). The Cox proportional hazards model demonstrated that SVR achievement was the only independent factor associated with low HCC recurrence (*p* = 0.005). In conclusion, patients who achieved SVR were the least prone to HCC recurrence, whereas those who did not achieve SVR and had AFP levels ≥ 11 ng/mL were the most prone to HCC recurrence.

**Keywords:** hepatocellular carcinoma; hepatitis C virus; recurrence risk; decision tree analysis; sustained virological response; alpha-fetoprotein

## 1. Introduction

Hepatocellular carcinoma (HCC) generally develops in patients with chronic liver damage due to various causative agents, such as persistent hepatitis B virus (HBV) and hepatitis C virus (HCV) infections, alcohol consumption, and obesity- and diabetes-related metabolic disorders [1]. Approximately 30% of the HCC cases worldwide occur due to HCV infection [2]. The mortality rate of HCC is relatively higher than that of other malignancies [3], partly because it can easily recur in regions other than the initial one, even if curative treatment was performed for the initial HCC. In fact, the 5-year recurrence rate after curative treatment is reportedly over 70% [3,4]. Thus, to reduce the mortality rate of HCC, it is essential to evaluate the risk factors of recurrence after curative treatment and consequently take proper precautions and/or improve HCC surveillance.

Several studies have suggested that the risk of HCC recurrence after curative treatment increases with male sex, presence of cirrhosis, large tumor foci, a multiplicity of tumors, pathologically high-grade atypia of tumor cells, presence of portal venous invasion, and high alpha-fetoprotein (AFP) levels [5–8]. Recently, obesity-related metabolic disorders,

such as diabetes mellitus (DM), have attracted attention as risk factors for the development and recurrence of HCC [9–12]. In addition, we previously demonstrated that HCC recurrence is promoted by high levels of homeostasis model assessment-insulin resistance (HOMA-IR), leptin, an adipocytokine secreted more in the obese, derivatives of reactive oxygen metabolites, a serum marker of oxidative stress, and excess accumulation of visceral adipose tissue [9–12].

In terms of HCV-related HCC, the eradication of HCV with direct-acting antivirals (DAA) is reported to suppress HCC development, similar to interferon (IFN)-based therapy [13,14]. However, some conflicting data have described unexpectedly high rates of HCC occurrence and recurrence following successful DAA therapy [15]. Therefore, it seems that various factors such as liver functional reserve, treatment for HCV and its components and response, progression of the initial HCC, and presence of metabolic disorders intricately influence each other in HCC recurrence after curative treatment in HCV-positive patients. However, the precise mechanism regarding the involvement of these factors in HCC recurrence remains unclear.

Among the possible risk factors described above, this study evaluated those that could affect HCC recurrence in HCV-positive patients after curative treatment using decision tree analysis.

## 2. Materials and Methods

### 2.1. Patients, Treatment, and Determination of HCC Recurrence

In total, 257 HCV-positive patients with HCC were treated at our hospital between May 2006 and December 2020. Among them, 23 patients who had achieved sustained virological response (SVR) before being diagnosed with the initial HCC, 106 patients who had not received curative treatment, and 17 patients who had local recurrence after curative treatment were excluded. Local recurrence was caused by inadequate treatment and was not regarded as "curative" because cancer cells must have survived despite treatment for the initial HCC. Thus, we believe that cases of local recurrence should be excluded from this study. The remaining 111 patients were enrolled in this study. The patient selection flow chart is shown in Figure 1.

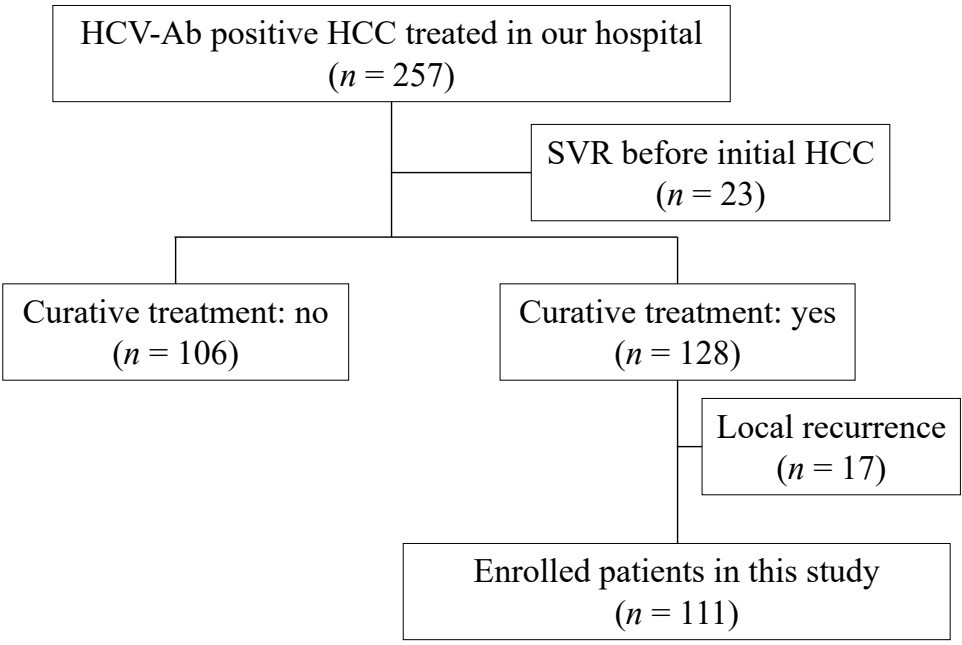

**Figure 1.** Patient selection flow chart.

HCC was diagnosed according to the clinical practice guidelines published by the Japan Society of Hepatology [16]. We diagnosed lesions visualized as high-attenuation

areas in the arterial phase and low-attenuation areas in the portal/equilibrium phase of dynamic computed tomography (CT) or magnetic resonance imaging (MRI) as typical HCC. If they did not meet the criteria for diagnosing HCC as described above, we conducted an additional CT during arterial portography, hepatic arteriography, or liver biopsy. Therapeutic effects were judged curative when surgical resection or radiofrequency ablation (RFA) was performed as a treatment for initial HCC, and dynamic studies showed the total disappearance of the imaging characteristics of HCC described above. After curative treatment, dynamic CT, MRI, or ultrasonography was performed every three months to detect recurrent HCC. Recurrent HCC was defined as a lesion that had the imaging characteristics of HCC in a different location from the initial HCC so as to exclude local recurrence.

This study did not require written informed consent from the participants as it was a retrospective and observational study that required only pre-existing samples or medical information. However, we provided the participants with an opportunity to opt out by disclosing the details of the study. The study design, including the consent procedure, was approved by the Ethics Committee of the Gifu University School of Medicine on 7 June 2017 (ethical protocol code: 29–26).

*2.2. Decision Tree Analysis of Risk Factors Affecting Recurrence of HCV-Related HCC*

The risk factors affecting HCC recurrence in HCV-positive patients who had received curative treatment for initial HCC were evaluated using decision tree analysis [17]. In this analysis, the recurrence-free survival data was the objective variable, and the following variables were the explanatory variables: I. Patient information, including age, sex, and SVR achievement after curative treatment. II. Body composition, including body mass index (BMI), presence of sarcopenia, subcutaneous adipose tissue index (SATI), and visceral adipose tissue index (VATI). III. Liver functional reserve, including the Child–Pugh score (CPS), platelet count (PLT), and Mac2 binding protein glycosylation isomer (M2BPGi). IV. Tumor factors, including AFP, proteins induced by vitamin K's absence or antagonist-II (PIVKA-II), cancer stage, and initial treatment (surgical resection or RFA). V. Metabolic syndrome, including the presence of DM, hyperlipidemia, and hypertension. VI. Insulin resistance, including fasting plasma glucose (FPG), fasting immunoreactive insulin (FIRI), HOMA-IR, and hemoglobin A1c (HbA1c). The study's outline using the decision tree analysis is shown in Figure 2.

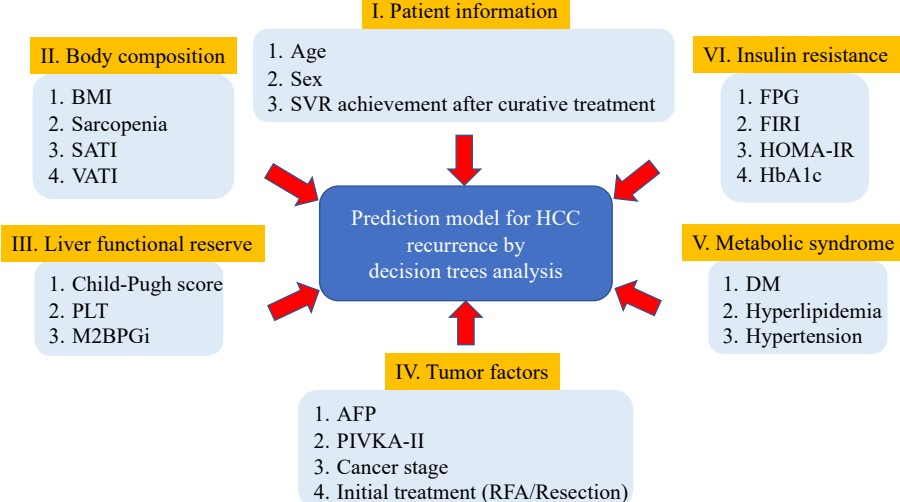

**Figure 2.** Study outline to evaluate the risk factors affecting hepatitis C virus-related recurrence of hepatocellular carcinoma after curative treatment using decision tree analysis.

Sarcopenia was defined based on a previous report [18]. The cross-sectional areas of the subcutaneous adipose tissue and visceral adipose tissue (cm$^2$) at the umbilical point

were normalized by the square of the height ($m^2$) to obtain the SATI and VATI ($cm^2/m^2$) values, respectively [11]. SYNAPSE VINCENT software (Fujifilm Medical, Tokyo, Japan) was used to measure the cross-sectional areas of these tissues.

### 2.3. Statistical Analyses

We conducted a decision tree analysis using the "rpart" package (version 4.1–15) in R under the following conditions: the method was "exp," the minimum number of observations was 30, the complexity parameter was 0.041, and the other settings were kept as their default values. Recurrence-free survival was estimated using the Kaplan–Meier method, and differences between the groups were compared using the log-rank test. We also used the Cox proportional hazards model to evaluate independent risk factors for HCC recurrence among those extracted using the decision tree analysis. Statistical significance was set at $p < 0.05$. All statistical analyses were performed using R, version 4.0.5 (R Foundation for Statistical Computing, Vienna, Austria; http://www.R-project.org/, accessed on 28 February 2022).

## 3. Results

### 3.1. Baseline Characteristics and Laboratory Data of the Participants

Table 1 shows the baseline characteristics and laboratory data of the 111 patients (73 males and 38 females; average age, 71.7 years) prior to the curative treatment for initial HCC. Regarding liver function reserve, 60, 30, 14, 5, 1, and 1 patients had CPS of 5, 6, 7, 8, 9, and 10, respectively. The average AFP and PIVKA-II levels were 417 ng/mL and 759 mAU/mL, respectively. As curative treatment, 41 patients underwent surgical resection and 70 underwent RFA. As antiviral therapy after curative treatment for initial HCC, 20 patients received DAA therapy (asunaprevir/daclatasvir, $n = 7$; sofosbuvir/ledipasvir, $n = 7$; ombitasvir/paritaprevir/ritonavir, $n = 3$; sofosbuvir/velpatasvir, $n = 2$; and glecaprevir/pibrentasvir, $n = 1$), and all of them achieved SVR. On the other hand, 17 patients received IFN therapy (pegylated-IFN alone, $n = 8$; pegylated-IFN/ribavirin, $n = 7$; and pegylated-IFN/ribavirin plus simeprevir, $n = 2$), of which 13 achieved SVR. We considered antiviral therapy after curative treatment after assessing its safety and efficacy. However, 64 patients did not receive any antiviral therapy during the observational period in this study because few patients refused taking treatment due to the adverse effect of the drug and the rest of them had recurrent HCC before the introduction of the therapy.

**Table 1.** Baseline demographic and clinical characteristics of the participants during initial hepatocellular carcinoma.

| Variables | ($n = 111$) |
|---|---|
| Sex (male/female) | 73/38 |
| Age (years) | $71.7 \pm 8.1$ |
| BMI (kg/m$^2$) | $22.3 \pm 3.0$ |
| Sarcopenia (yes/no) | 44/67 |
| SATI (cm$^2$/m$^2$) | $36.5 \pm 22.3$ |
| VATI (cm$^2$/m$^2$) | $33.1 \pm 23.2$ |
| Diabetes mellitus (yes/no) | 32/79 |
| Hypertension (yes/no) | 42/69 |
| Hyperlipidemia (yes/no) | 1/110 |
| FPG (mg/dL) | $107.6 \pm 29.6$ |
| FIRI (µIU/mL) | $10.9 \pm 9.7$ |
| HOMA-IR | $2.9 \pm 3.1$ |
| HbA1c (%) | $5.9 \pm 1.1$ |
| Child-Pugh score (5/6/7/8/9/10) | 60/30/14/5/1/1 |
| M2BPGi (C.O.I) | $5.0 \pm 3.6$ |
| PLT ($\times 10^4$/µL) | $11.7 \pm 5.5$ |
| Stage (I/II/III/IV) | 50/47/13/1 |
| AFP (ng/mL) | $417 \pm 2039$ |
| PIVKA-II (mAU/mL) | $759 \pm 3268$ |
| Initial treatment (resection/RFA) | 41/70 |
| SVR achievement after curative treatment (yes/no) | 33/78 |

Values are presented as means $\pm$ standard deviations or numbers. BMI, body mass index; SATI, subcutaneous adipose tissue index; VATI, visceral adipose tissue index; FPG, fasting plasma glucose; FIRI, fasting immunoreactive insulin; HOMA-IR, homeostasis model assessment-insulin resistance; HbA1c, hemoglobin A1c; M2BPGi, Mac2 binding protein glycosylation isomer; PLT, platelet count; AFP, alpha-fetoprotein; PIVKA-II, protein induced by vitamin K absence or antagonist-II; RFA, radiofrequency ablation; SVR, sustained virological response.

### 3.2. Risk Factors Affecting HCC Recurrence in HCV-Positive Patients after Curative Treatment

The results of the decision tree analysis are shown in Figure 3. Patients who achieved SVR after curative treatment (*n* = 33) had the longest recurrence-free survival. Furthermore, among the patients who did not achieve SVR (*n* = 78), those with AFP levels ≥ 11 ng/mL (*n* = 48) had poorer recurrence-free survival than those with AFP levels < 11 ng/mL (*n* = 30). Table 2 shows the baseline demographic breakdown divided into these three groups (see Figure 3 for grouping), and there were no significant differences in any variables except recurrence-free survival (RFS) ($p$ < 0.001).

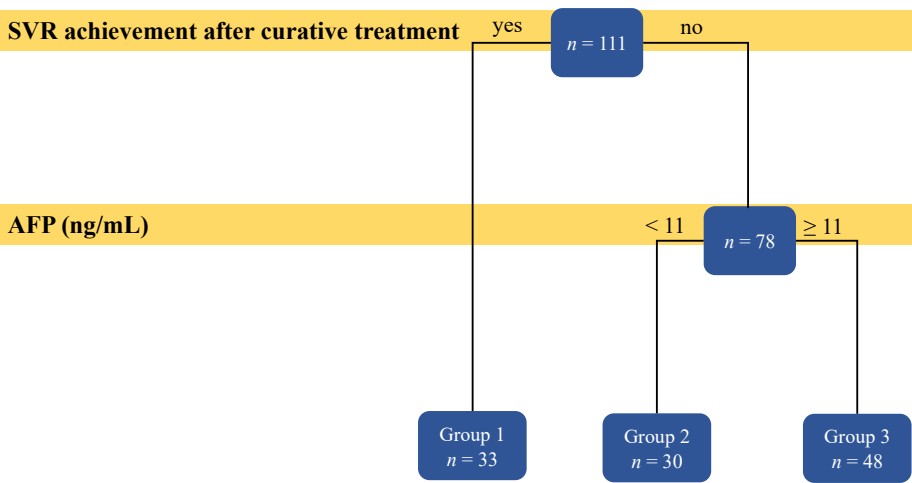

**Figure 3.** Result of the decision tree analysis on the factors predicting hepatitis C virus-related recurrence of hepatocellular carcinoma.

**Table 2.** Baseline demographic and clinical characteristics are divided into Groups 1, 2, and 3.

| Variables | Group 1 | Group 2 | Group 3 | *p*-Value |
|---|---|---|---|---|
| Sex (male/female) | 24/9 | 22/8 | 27/21 | 0.210 |
| Age (years) | 70.4 ± 7.4 | 74.4 ± 8.4 | 70.8 ± 8.1 | 0.092 |
| BMI (kg/m$^2$) | 21.5 ± 3.0 | 22.5 ± 3.3 | 22.7 ± 2.8 | 0.240 |
| Sarcopenia (yes/no) | 12/21 | 13/17 | 19/29 | 0.827 |
| SATI (cm$^2$/m$^2$) | 32.9 ± 23.7 | 35.3 ± 23.7 | 39.7 ± 20.2 | 0.374 |
| VATI (cm$^2$/m$^2$) | 28.8 ± 20.5 | 36.0 ± 26.5 | 34.1 ± 22.8 | 0.432 |
| Diabetes mellitus (yes/no) | 9/24 | 9/21 | 14/34 | 1.000 |
| Hypertension (yes/no) | 18/15 | 10/20 | 14/34 | 0.069 |
| Hyperlipidemia (yes/no) | 0/33 | 0/30 | 1/47 | 1.000 |
| FPG (mg/dL) | 108.7 ± 23.5 | 110.2 ± 45.9 | 105.2 ± 19.8 | 0.753 |
| FIRI (μIU/mL) | 12.5 ± 11.1 | 9.8 ± 11.7 | 10.5 ± 7.1 | 0.558 |
| HOMA-IR | 3.4 ± 3.1 | 2.7 ± 4.5 | 2.8 ± 2.0 | 0.665 |
| HbA1c (%) | 6.0 ± 1.1 | 6.0 ± 1.5 | 5.8 ± 0.8 | 0.563 |
| Child-Pugh score (5/6/7/8/9/10) | 18/9/5/1/00 | 19/5/4/1/0/1 | 23/16/5/3/1/0 | 0.741 |
| M2BPGi (C.O.I) | 5.1 ± 5.1 | 4.4 ± 2.8 | 5.3 ± 2.9 | 0.880 |
| PLT (×10$^4$/μL) | 12.4 ± 5.4 | 13.5 ± 5.9 | 10.1 ± 4.9 | 0.211 |
| Stage (I/II/III/IV) | 15/16/2/0 | 15/11/3/1 | 20/20/8/0 | 0.560 |
| AFP (ng/mL) | 288.6 ± 727.8 | 4.9 ± 2.7 | 763.9 ± 3022.6 | 0.256 |
| PIVKA-II (mAU/mL) | 1137.9 ± 5026.9 | 385.2 ± 1297.2 | 734.1 ± 2305.2 | 0.662 |
| Initial treatment (resection/RFA) | 15/18 | 11/19 | 15/33 | 0.672 |
| Median RFS (days) | 2009 | 1267 | 501 | <0.001 |

Values are presented as mean ± standard deviation or number. BMI, body mass index; SATI, subcutaneous adipose tissue index; VATI, visceral adipose tissue index; FPG, fasting plasma glucose; FIRI, fasting immunoreactive insulin; HOMA-IR, homeostasis model assessment-insulin resistance; HbA1c, hemoglobin A1c; M2BPGi, Mac2 binding protein glycosylation isomer; PLT, platelet count; AFP, alpha-fetoprotein; PIVKA-II, protein induced by vitamin K absence or antagonist-II; RFA, radiofrequency ablation; RFS, recurrence-free survival.

The 1-year, 3-year, and 5-year RFS rates of the participants were 84.0%, 42.0%, and 28.3%, respectively, and the median RFS was 882 days (Figure 4a). Patients who achieved

SVR after curative treatment had a significantly longer recurrence-free survival than those who did not ($p$ = 0.004; Figure 4b). However, there was no significant difference in the RFS between patients who achieved SVR with DAA and IFN therapy ($p$ = 0.251; Figure 4c).

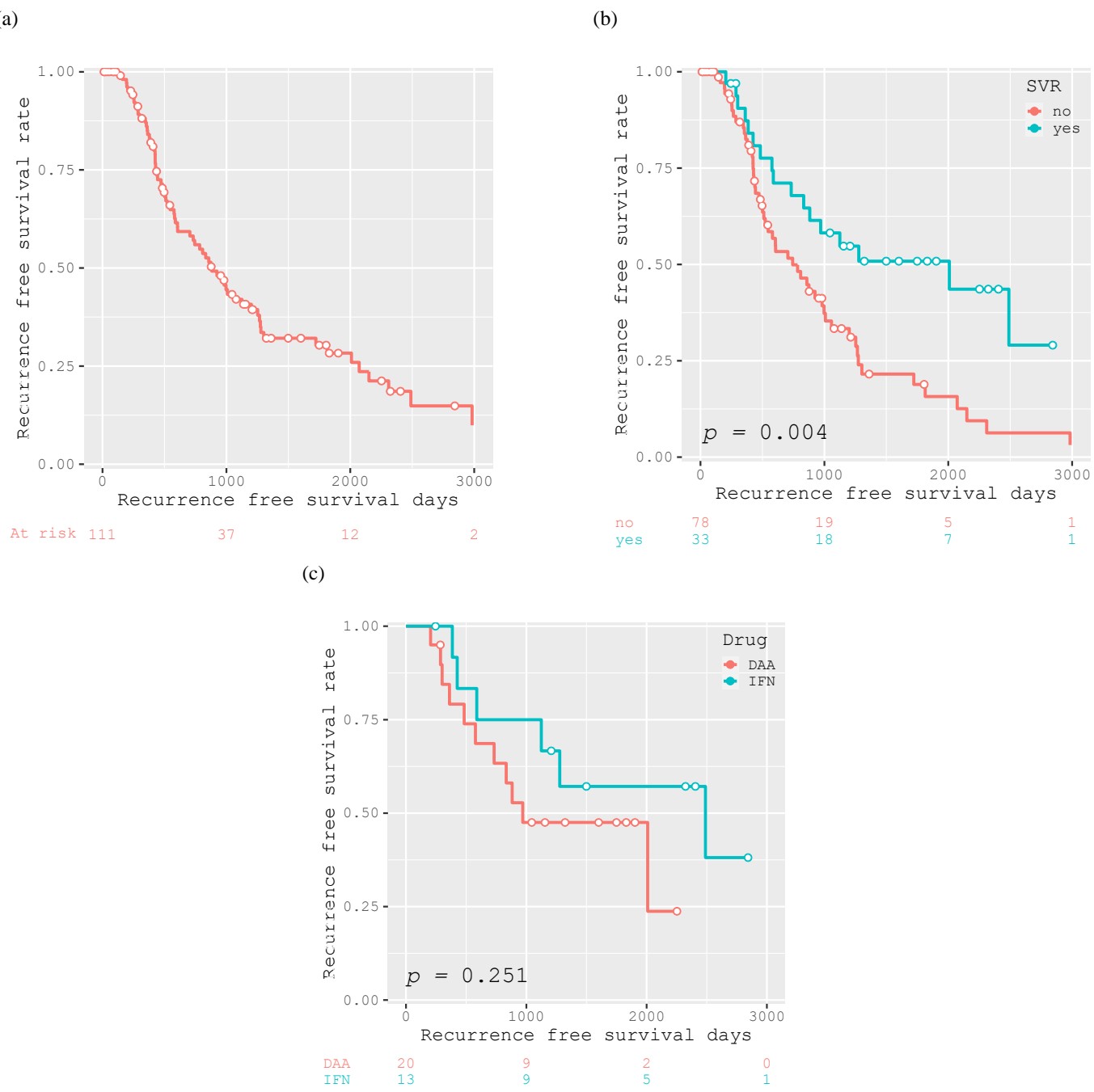

**Figure 4.** Kaplan–Meier curves for recurrence-free survival after curative treatment for all participants ($n$ = 111) (**a**), for participants divided into two groups according to SVR achievement ($n$ = 33) or not ($n$ = 78) (**b**), and for the 33 participants achieving SVR who were divided according to antiviral therapy with DAA ($n$ =20) or IFN ($n$ =13) for HCV (**c**).

Among the patients who achieved SVR, there was no significant difference in the RFS rate between patients with AFP levels ≥ 11 ng/mL and <11 ng/mL ($p$ = 0.362; Figure 5a). On the other hand, among the patients who did not achieve SVR, those with AFP levels ≥ 11 ng/mL had significantly poorer RFS than those with AFP levels < 11 ng/mL ($p$ < 0.001; Figure 5b).

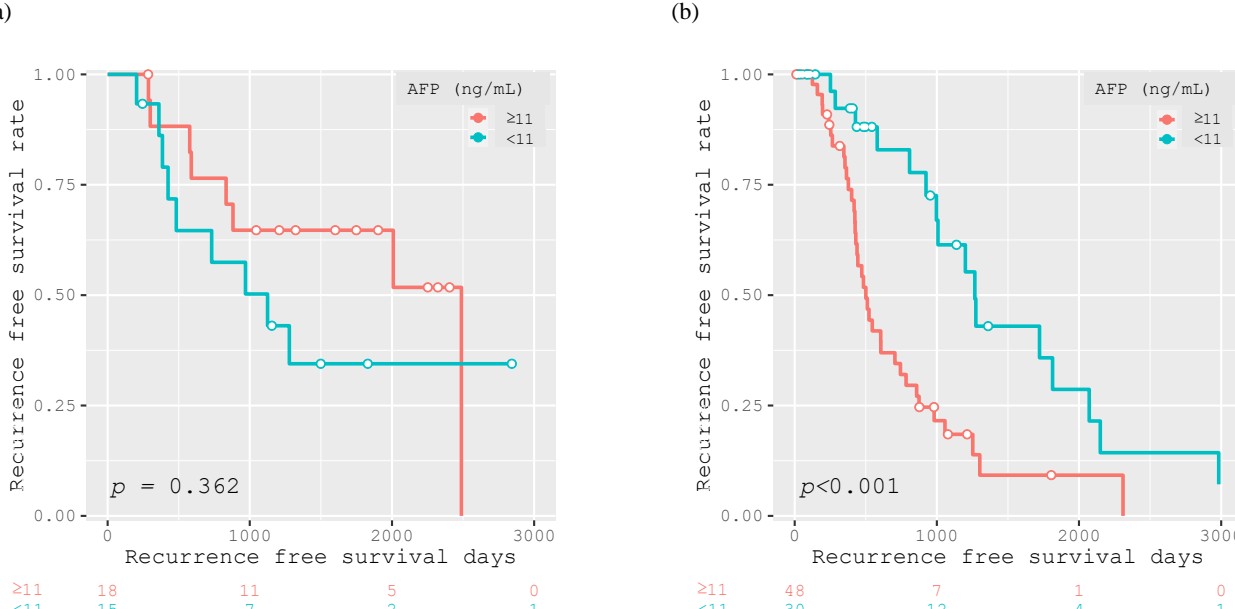

**Figure 5.** Kaplan–Meier curves for recurrence-free survival after curative treatment for 33 participants achieving SVR (AFP ≥ 11 ng/mL, *n* = 18; AFP < 11 ng/mL, *n* = 15) (**a**) and for 78 participants without SVR (AFP ≥ 11 ng/mL, *n* = 48; AFP < 11 ng/mL, *n* = 30) (**b**), with group division based on the AFP cut off value of 11 ng/mL.

The Cox proportional hazards model demonstrated that SVR achievement after curative treatment was the only independent factor suppressing HCC recurrence (hazard ratio: 0.449, 95% confidence interval: 0.257–0.783; *p* = 0.005) (Table 3).

**Table 3.** Analysis of the possible risk factors for hepatocellular carcinoma recurrence using the Cox proportional hazards model.

| Variables | HR (95% CI) | *p*-Value |
|---|---|---|
| Sex (male vs. female) | 0.978 (0.594–1.611) | 0.932 |
| Age (years) | 0.999 (0.965–1.035) | 0.971 |
| BMI (kg/m$^2$) | 1.028 (0.950–1.112) | 0.487 |
| Sarcopenia (yes vs. no) | 0.739 (0.448–1.220) | 0.238 |
| SATI (cm$^2$/m$^2$) | 1.004 (0.993–1.015) | 0.428 |
| VATI (cm$^2$/m$^2$) | 1.005 (0.994–1.015) | 0.352 |
| Diabetes mellitus (yes vs. no) | 0.853 (0.493–1.477) | 0.572 |
| Hypertension (yes vs. no) | 0.930 (0.571–1.516) | 0.773 |
| Hyperlipidemia (yes vs. no) | 2.368 (0.323–17.33) | 0.396 |
| FPG (mg/dL) | 0.994 (0.983–1.004) | 0.246 |
| FIRI (μIU/mL) | 1.009 (0.989–1.029) | 0.365 |
| HOMA-IR | 1.022 (0.961–1.087) | 0.491 |
| HbA1c (%) | 0.810 (0.620–1.059) | 0.124 |
| Child-Pugh score | 1.042 (0.794–1.367) | 0.766 |
| M2BPGi (C.O.I) | 0.932(0.778–1.117) | 0.447 |
| PLT (×10$^4$/μL) | 0.962 (0.918–1.008) | 0.105 |
| Stage | 1.365 (0.948–1.965) | 0.093 |
| AFP (≥11 vs. <11 ng/mL) | 1.619 (0.981–2.673) | 0.059 |
| Initial treatment (RFA vs. resection) | 1.166 (0.708–1.917) | 0.546 |
| SVR achievement (yes vs. no) | 0.449 (0.257–0.783) | 0.005 |

HR, hazard ratio; CI, confidence interval; BMI, body mass index; SATI, subcutaneous adipose tissue index; VATI, visceral adipose tissue index; FPG, fasting plasma glucose; FIRI, fasting immunoreactive insulin; HOMA-IR, homeostasis model assessment-insulin resistance; HbA1c, hemoglobin A1c; M2BPGi, Mac2 binding protein glycosylation isomer; PLT, platelet count; AFP, alpha-fetoprotein; RFA, radiofrequency ablation; SVR, sustained virological response.

## 4. Discussion

This study clearly demonstrated that SVR achievement after curative treatment for initial HCC had the strongest inhibitory effect on HCC recurrence. Furthermore, SVR achievement was the only predictor of HCC recurrence, with a reduction in the risk of HCC recurrence by more than 50% in this study. These findings are consistent with previous reports showing that achieving SVR is critically important in reducing the recurrence of HCV-related HCC, regardless of IFN-based or DAA therapies [13,14]. To the best of our knowledge, this is the first study to evaluate the risk factors that would affect HCC recurrence in HCV-positive patients who had received curative treatment for initial HCC using decision tree analysis, which revealed that SVR achievement was the most suppressive factor for HCC recurrence among all the possible risk factors for HCC recurrence, including liver functional reserve, progression of the initial HCC, and presence of metabolic disorders [5–12,19].

Recently, DAA therapy has completely replaced the conventional IFN-based therapy in the field of antiviral HCV therapy, because of its comparatively higher tolerability and SVR rate [20]. Immediately after the advent of DAA therapy, there were some concerns suggesting an unexpectedly high rate of early tumor recurrence in patients with HCV-related HCC undergoing DAA therapy [15]. However, several studies have demonstrated that there is no evidence of differential HCC recurrence risk following SVR with DAA- and IFN-based therapies [13,14], which seems to be the predominant standpoint currently; the results of the present study did not contradict this finding. Elderly patients and patients with more advanced cirrhosis, both of whom are at a high risk of HCC [1], are eligible for DAA therapy. The efficacy and safety of DAA treatment in patients with decompensated cirrhosis have been established [21,22]. Thus, we should consider the introduction of DAA therapy for any HCV-positive patient, who had received curative treatment for initial HCC, to ensure prevention of recurrence as well as improvement of liver function and quality of life [21,23,24].

The decision tree analysis conducted in this study also demonstrated that patients who did not achieve SVR and had AFP levels $\geq 11$ ng/mL were most prone to HCC recurrence. High AFP levels reportedly indicate severe vascular invasion and the existence of minute lesions that cannot be detected using imaging modalities [6]. AFP > 400–500 ng/mL is considered diagnostic for HCC, although fewer than half of patients may generate levels that high, suggesting that an AFP level that is negative or below 400 ng/mL should not exclude HCC [25]. Although AFP must be used with imaging modalities in HCC screening, AFP could be useful in predicting the recurrence risk of HCC after curative treatment in patients who did not achieve SVR. However, those who achieved SVR with lower AFP levels did not have more prolonged recurrence-free survival (Figure 5a), which is at least one of the reasons why AFP was not chosen as an independent risk factor for HCC recurrence using the Cox proportional hazards model. This may be due to the small sample size and short observation period or for reasons that we do not know now yet. A future challenge is to clarify whether AFP could be useful in predicting recurrence risk in patients who achieve SVR. According to the HCC guidelines in Japan [16], patients who received curative treatment for initial HCC were classified as the "extremely high-risk group" due to their high recurrence rate. Within this "extremely high-risk" group, those who achieved SVR were the least prone to recurrent HCC, whereas those who did not achieve SVR and had AFP levels $\geq 11$ ng/mL were the most prone to recurrent HCC. These facts will be beneficial in a clinical setting when we follow up patients who have received curative treatment for their initial HCC. For instance, the strictest HCC surveillance, such as dynamic CT or MRI imaging and measurement of tumor markers, should be performed for those who do not achieve SVR and have higher AFP levels at least every three months.

Contrary to the expectations, none of the recurrence risk factors involved in liver functional reserve and metabolic disorders reported in previous studies [5–12] were selected as significant recurrence risk factors in this study. This may be partly because SVR achievement can affect HCC recurrence substantially and cancel the effect of other possible

risk factors for HCC recurrence and partly because the sample size of this study was too small for the other possible risk factors to be defined as significant. This is one of the limitations of this study; therefore, additional analysis of a higher number of cases is warranted to clarify the involvement of liver functional reserve and metabolism-related factors in HCC recurrence.

Another limitation is that this was a retrospective, single-center study, with the potential for bias in patient enrollment. HCC is diagnosed mainly by imaging modalities, such as CT or MRI, and recurrent HCC cannot be strictly classified as multicentric occurrence or intrahepatic metastasis. Therefore, a prospective study involving a higher number of HCV-positive patients with HCC enrolled from several centers is warranted.

## 5. Conclusions

Among the HCV-positive patients who had received curative treatment for initial HCC, those who achieved SVR after curative treatment were the least prone to HCC recurrence. On the other hand, those who did not achieve SVR and had AFP levels $\geq 11$ ng/mL prior to curative treatment were the most prone to HCC recurrence. Achieving SVR is crucial in the prevention of HCV-related HCC, and strict surveillance for HCC should be conducted in cases without SVR with AFP levels $\geq 11$ ng/mL.

**Author Contributions:** K.I., K.T., S.U., T.M. (Takao Miwa), T.M. (Toshihide Maeda), T.H., Y.S., A.S. and M.S. designed the study. K.I. analyzed the data and drafted the manuscript. K.T. supervised the participants' treatment. K.T., S.U., T.M. (Takao Miwa), T.M. (Toshihide Maeda), T.H., Y.S. and A.S. contributed to participant selection and data collection. K.T., S.U., T.M. (Takao Miwa), T.M. (Toshihide Maeda), T.H., Y.S. and A.S. revised the manuscript, and M.S. reviewed and amended the manuscript. All authors have read and agreed to the published version of the manuscript.

**Funding:** This study received no external funding.

**Institutional Review Board Statement:** The study design, including the consent procedure, was approved by the Ethics Committee of Gifu University School of Medicine (ethical protocol code: 29–26, approved on 7 June 2017).

**Informed Consent Statement:** We were unable to obtain written informed consent in advance due to the retrospective design of our study. Instead, by disclosing the details of the study, we provided the participants an opportunity to opt out.

**Data Availability Statement:** The data presented in this study are available upon reasonable request from the corresponding author.

**Conflicts of Interest:** The authors have no conflict of interest to declare.

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
