# Peer review of "Sustained Virological Response Is the Most Effective in Preventing Hepatocellular Carcinoma Recurrence after Curative Treatment in Hepatitis C Virus-Positive Patients: A Study Using Decision Tree Analysis"

_2673-8937, doi:10.3390/ijtm2030027_

Round 1

Reviewer 1 Report

The manuscript by Kenji Imai et al. entitled “Sustained virological response is the most effective in preventing hepatocellular carcinoma recurrence after curative treatment in hepatitis C virus-positive patients: a study using decision tree analysis” is an excellent evaluation of risk factors associated with hepatocellular carcinoma recurrence. The authors are well versed in the field and demonstrated that patients that achieved SVR exhibited the be best indicator of recurrence-free survival. For the patients that did not achieve SVR, elevated levels of AFP showed a poorer recurrence- free survival. These data are very important and could be utilized to identify patients at increased risk for recurrence. The limitations of this study are a relatively small cohort and a single site study, but both limitations are clearly addressed in the discussion. The only item that would be suggested to strengthen the findings would be a baseline demographics breakdown of the different groups (SVR yes/no, AFP >11ng/mL<). This may help inform any additional trends that may be within the groups. 

Author Response

July 14, 2022

Prof. Dr. Pier Paolo Claudio

Editor-in-Chief

International Journal of Translational Medicine

Re: Revision of International Journal of Translational Medicine

Manuscript ID: ijtm-1803465

Title: Sustained virological response is the most effective in preventing hepatocellular carcinoma recurrence after curative treatment in hepatitis C virus-positive patients: a study using decision tree analysis

Dear Prof. Dr. Pier Paolo Claudio

Thank you for your mail of July 6, 2022 regarding the above manuscript and for the reviewers’ comments. We are pleased to learn that you have been interested in our manuscript. We found the raised concerns were all valid and also very suggestive to strengthen our study. We deeply thank you and the reviewers for these constructive comments. Accordingly, we revised the whole manuscript extensively.

Please find the revised manuscript, in which revisions are marked in red and point-by-point responses to the reviewers’ comments. We believe the modifications fulfill the requirements made by the reviewers and would like to request re-evaluation of the manuscript for possible publication in International Journal of Translational Medicine.

Sincerely,

Kenji Imai MD, phD

Department of Gastroenterology/Internal Medicine, Gifu University Graduate School of Medicine 1-1 Yanagido, Gifu 501-1194, Japan

Tel.: +81-(58)-230-6308

Fax: +81-(58)-230-6310

Email: ikenji@gifu-u.ac.jp

Response to Reviewer #1

We are pleased that in the overall comments this reviewer evaluated our study very highly. We also thank this reviewer’s constructive comments which were most helpful to improve our manuscript. We accordingly revised the manuscript as follows.

  1. The only item that would be suggested to strengthen the findings would be a baseline demographics breakdown of the different groups (SVR yes/no, AFP >11ng/mL<). This may help inform any additional trends that may be within the groups.

According to this suggestion, we added a new Table 2. There were no significant differences in any variables except for recurrence free survival among them (lines 160-162 and 166-172).

Reviewer 2 Report

Significance of the present study and the authors previous article in “Mol Clin Oncol. 2020; 12(2): 111–116” are sustained virological response is the most effective in preventing HCC recurrence for longer time. And in both the studies SVR is achieved in patients who are under DAA or IFN based therapy. In addition, Authors mentioned “SVR achievement was the only predictor of HCC recurrence, with a reduction in the risk of HCC recurrence by more than 50% in this study”. Hence, I don’t find any novelty in the present study.

What is the rationale behind in treating fewer patients with DAA and IFN and rest of them were not?

Authors mentioned “17 patients who had local recurrence after curative treatment were excluded.” Explain what is local recurrence and how these data impact the present study?

Authors previous study “Mol Clin Oncol. 2020; 12(2): 111–116”, between 2006-2017 had 34 patients who administered IFN based therapy. However, in the present study, between 2006-2020 had data only on 17 patients who administered IFN based therapy. It looks like the data were collected from the same hospital. Could you please explain the inclusion and exclusion criteria in detail?

Based on the AFP values and analysis presented in this paper, Decision tree analysis demonstrated AFP was one of the factors in predicting HCC recurrence and Cox model doesn’t. However, around 30 HCC patients out of 78 (who did not achieve SVR) had AFP <11 ng/ml before curative treatment. This is in contrast with the literatures that AFP >400–500 ng/ml is considered diagnostic for HCC. How sensitive is the decision tree analysis in predicting HCC/ recurrence of HCC? How many patients had the AFP between 11 and 400 ng/ml?

In addition, irrespective of AFP values, SVR achieved group had longer recurrence free survival (figure 5a). Hence, AFP alone does not consider as predictor for the determination of HCC recurrence? If the SVR is achieved in 78 patients, the outcome from decision tree analysis based on AFP values might be different.

What are the AFP values of Group 1 (SVR achieved after curative treatment)?

What are the average days for recurrence of HCC in two different groups ie >11 and <11 AFP group?

Author Response

July 14, 2022

Prof. Dr. Pier Paolo Claudio

Editor-in-Chief

International Journal of Translational Medicine

Re: Revision of International Journal of Translational Medicine

Manuscript ID: ijtm-1803465

Title: Sustained virological response is the most effective in preventing hepatocellular carcinoma recurrence after curative treatment in hepatitis C virus-positive patients: a study using decision tree analysis

Dear Prof. Dr. Pier Paolo Claudio

Thank you for your mail of July 6, 2022 regarding the above manuscript and for the reviewers’ comments. We are pleased to learn that you have been interested in our manuscript. We found the raised concerns were all valid and also very suggestive to strengthen our study. We deeply thank you and the reviewers for these constructive comments. Accordingly, we revised the whole manuscript extensively.

Please find the revised manuscript, in which revisions are marked in red and point-by-point responses to the reviewers’ comments. We believe the modifications fulfill the requirements made by the reviewers and would like to request re-evaluation of the manuscript for possible publication in International Journal of Translational Medicine.

Sincerely,

Kenji Imai MD, phD

Department of Gastroenterology/Internal Medicine, Gifu University Graduate School of Medicine 1-1 Yanagido, Gifu 501-1194, Japan

Tel.: +81-(58)-230-6308

Fax: +81-(58)-230-6310

Email: ikenji@gifu-u.ac.jp

Response to Reviewer #2

We are pleased that in the overall comments this reviewer found our study is of interest. We also thank this reviewer’s constructive comments which were most helpful to improve our manuscript. We accordingly revised the manuscript as follows.

  1. Significance of the present study and the authors previous article in “Mol Clin Oncol. 2020; 12(2): 111–116” are sustained virological response is the most effective in preventing HCC recurrence for longer time. And in both the studies SVR is achieved in patients who are under DAA or IFN based therapy. In addition, Authors mentioned “SVR achievement was the only predictor of HCC recurrence, with a reduction in the risk of HCC recurrence by more than 50% in this study”. Hence, I don’t find any novelty in the present study.

In our previous article (Mol Clin Oncol. 2020; 12(2): 111–116), we focused on whether SVR achievement would suppress a recurrence risk for HCC after curative treatment from the start. On the other hand, in this study, we analyzed which factors would affect HCC recurrence the most among the established risk factors such as liver functional reserve, progression of the initial HCC, and presence of metabolic disorders in addition to the SVR achievement. As a result, the decision tree analysis revealed SVR achievement was the most suppressive factor for HCC recurrence. Furthermore, we also demonstrated that those who did not achieve SVR and had AFP levels ≥11 ng/mL prior to curative treatment were the most prone to HCC recurrence. We believe that these two facts must be extremely beneficial in a clinical setting when we follow up patients who received curative treatment for initial HCC. We emphasized the novelty in this study in the Discussion section (lines 214-215, 248-252, and 253-254).

  1. What is the rationale behind in treating fewer patients with DAA and IFN and rest of them were not?

We considered the introduction of anti-viral therapy in any cases. However, particularly IFN-based therapy can induce such serious adverse events as thrombocytopenia, liver functional impairment, fever, and general fatigue, and a considerable number of patients made a denial of antiviral therapy for fear of these adverse events. Furthermore, some patients had recurrent HCC before the introduction of antiviral therapy. Eventually, 64 patients did not receive any anti-viral therapy at all during the observational period in this study. We described the details in the criteria of the anti-viral therapy after curative treatment for initial HCC (lines 142-146).

  1. Authors mentioned “17 patients who had local recurrence after curative treatment were excluded.” Explain what is local recurrence and how these data impact the present study?

Local recurrence was caused by an insufficient treatment and was not regarded as “curative” because cancer cells must have survived just after the treatment for initial HCC. Thus, we think that local recurrence cases should be excluded in this study. We added the reason why we excluded local recurrence cases more clearly (lines 70-73).

  1. Authors previous study “Mol Clin Oncol. 2020; 12(2): 111–116”, between 2006-2017 had 34 patients who administered IFN based therapy. However, in the present study, between 2006-2020 had data only on 17 patients who administered IFN based therapy. It looks like the data were collected from the same hospital. Could you please explain the inclusion and exclusion criteria in detail?

Local recurrence and SVR achievement before initial treatment cases were excluded beforehand in this study, both of which were included in our previous article (Mol Clin Oncol. 2020; 12(2): 111–116). Thus, although the number of patients who administrated IFN-based therapy looks like decreased in this study, the criteria of the anti-viral therapy are the same between our previous and present study. We described the details in the criteria of the antiviral therapy after curative treatment for initial HCC (lines 142-146).

  1. Based on the AFP values and analysis presented in this paper, Decision tree analysis demonstrated AFP was one of the factors in predicting HCC recurrence and Cox model doesn’t. However, around 30 HCC patients out of 78 (who did not achieve SVR) had AFP <11 ng/ml before curative treatment. This is in contrast with the literatures that AFP >400–500 ng/ml is considered diagnostic for HCC. How sensitive is the decision tree analysis in predicting HCC/ recurrence of HCC? How many patients had the AFP between 11 and 400 ng/ml?

The literature this reviewer mentioned (2001 Feb;5(1):109-22) also said that less than half of patients may generate levels that high. This fact suggests that an AFP that is negative or below 400 ng/ml of AFP should not exclude HCC. In fact, among those who did not achieve SVR (n = 78), the number of patients with 1-10, 11-400, and > 400 ng/ml of AFP were 30, 40, and 8, respectively. Thus, AFP test must be used together with imaging modalities in screening HCC. The decision tree analysis conducted in this study suggests that AFP could be useful in predicting recurrence risk of HCC after curative treatment in those who did not achieve SVR. We added this information about AFP in the Discussion section with a new reference #26 (lines 235-240).

  1. In addition, irrespective of AFP values, SVR achieved group had longer recurrence free survival (figure 5a). Hence, AFP alone does not consider as predictor for the determination of HCC recurrence? If the SVR is achieved in 78 patients, the outcome from decision tree analysis based on AFP values might be different.

We cannot tell in detail the reason why SVR achieved group with lower AFP level did not have longer recurrence free survival, which is at least one of the reasons why AFP was not chosen as an independent risk factor for HCC recurrence by Cox proportional hazards model. This may be due to the small sample size and short observation period or for reasons that we do not know now yet. We raised this point as a challenge for the future in the Discussion section (lines 240-246)

  1. What are the AFP values of Group 1 (SVR achieved after curative treatment)?

We added a new Table 2 that shows baseline demographic breakdown divided into the three groups. and there were no significant differences in any variables except recurrence free survival among them. Mean AFP levels of the Group 1, 2, and 3 were 288.6, 4.9, and 763.9 ng/ml, respectively. We added this description and a new Table 2 in the Results section (lines 160-162, and 166-172)

  1. What are the average days for recurrence of HCC in two different groups ie >11 and <11 AFP group?

Median recurrence free survival of the Group1, 2, and 3 were 2009, 1267, and 501 days, respectively. This result was represented in a new Table 2 (lines 166-172)

Round 2

Reviewer 2 Report

I thank authors for the response to the comments and revising the manuscript accordingly. I have few more minor comments appended below.

Even though the outcomes are from different analysis, Statements in the abstract looks duplicate. “The results showed that patients who achieved sustained virological response (SVR) after curative treatment (n = 33) had the longest recurrence-free survival. Furthermore, among the patients who did not achieve SVR (n = 78), those with alpha-fetoprotein (AFP) levels ≥11 ng/mL (n = 48) had a poor recurrence-free survival” and “The Kaplan–Meier method revealed that patients who achieved SVR had significantly longer recurrence-free survival than those who did not (p = 0.004). Among patients who did not achieve SVR, those with AFP levels ≥11 ng/mL had significantly poorer recurrence-free survival than those with AFP levels <11 ng/mL (p < 0.001).” Rewrite the abstracts accordingly.

Rewrite the sentence in ine 144-146 as “However, 74 patients (check the number) did not receive any antiviral therapy during the observational period in this study because few patients refused taking treatment due to the adverse effect of the drug and rest of them had recurrent HCC before the introduction of the therapy”

Authors have mentioned, “those with AFP levels ≥11 ng/mL (n = 48) had poorer recurrence-free survival than those with AFP levels <11 ng/mL (n = 30).” The question will raise whether the AFP level =11 falls in poor or longer HCC recurrence. Hence, correct the symbol, ≥ as > in throughout the manuscript.

Author Response

Response to Reviewer #2 

We are pleased that in the overall comments this reviewer found our study is of interest. We also thank this reviewer’s constructive comments which were most helpful to improve our manuscript. We accordingly revised the manuscript as follows.

  1. Even though the outcomes are from different analysis, Statements in the abstract looks duplicate. “The results showed that patients who achieved sustained virological response (SVR) after curative treatment (n = 33) had the longest recurrence-free survival. Furthermore, among the patients who did not achieve SVR (n = 78), those with alpha-fetoprotein (AFP) levels ≥11 ng/mL (n = 48) had a poor recurrence-free survival” and “The Kaplan–Meier method revealed that patients who achieved SVR had significantly longer recurrence-free survival than those who did not (p = 0.004). Among patients who did not achieve SVR, those with AFP levels ≥11 ng/mL had significantly poorer recurrence-free survival than those with AFP levels <11 ng/mL (p < 0.001).” Rewrite the abstracts accordingly.

According to this suggestion, we rewrote the abstract (lines 15-23).

  1. Rewrite the sentence in ine 144-146 as “However, 74 patients (check the number) did not receive any antiviral therapy during the observational period in this study because few patients refused taking treatment due to the adverse effect of the drug and rest of them had recurrent HCC before the introduction of the therapy”

We rewrote the sentence as you suggested (lines 145-146).

  1. Authors have mentioned, “those with AFP levels ≥11 ng/mL (n = 48) had poorer recurrence-free survival than those with AFP levels <11 ng/mL (n = 30).” The question will raise whether the AFP level =11 falls in poor or longer HCC recurrence. Hence, correct the symbol, ≥ as > in throughout the manuscript.

AFP level = 11 falls in the poorer RFS group (Figure 3). We changed the symbol “>” into “≥” in the line 26 and 275. We appreciate all your kind suggestions.